

# Limited effects of source population identity and number on seagrass transplant performance

Alyssa B. Novak[1], Holly K. Plaisted[2], Cynthia G. Hays[3] and Randall A. Hughes[4]

[1] Department of Earth and Environment, Boston University, Boston, MA, USA
[2] Cape Cod National Seashore, National Park Service, Wellfleet, MA, USA
[3] Department of Biology, Keene State College, Keene, NH, USA
[4] Marine Science Center, Northeastern University, Nahant, MA, USA

Corresponding author
Alyssa B. Novak, abnovak@bu.edu

## ABSTRACT

Global declines in coastal foundation species highlight the importance of effective restoration. In this study, we examined the effects of source population identity and diversity (one vs. three sources per plot) on seagrass (*Zostera marina*) transplant success. The field experiment was replicated at two locations in Massachusetts with adjacent natural *Zostera marina* beds to test for local adaptation and source diversity effects on shoot density. We also collected morphological and genetic data to characterize variation within and among source populations, and evaluate whether they were related to performance. Transplants grew and expanded until six months post-transplantation, but then steadily declined at both sites. Prior to declines, we observed variation in performance among source populations at one site that was related to morphological traits: the populations with the longest leaves had the highest shoot densities, whereas the population with the shortest leaves performed the worst at six months post-transplantation. In addition, multiple source plots at this same transplant site consistently had similar or higher shoot densities than single source plots, and shoots from weak-performing populations showed improved performance in multiple source plots. We found no evidence for home site advantage or benefits of population-level genetic variation in early transplant performance at either site. Our results show limited effects of source population on early transplant performance and suggest that factors (e.g., morphology) other than home site advantage and population genetic variation serve a role. Based on our overall findings that transplant success varied among source populations and that population diversity at the plot level had positive but limited effects on individual and plot performance, we support planting shoots from multiple source sites in combination to enhance transplant success, particularly in the absence of detailed information on individual source characteristics.

## INTRODUCTION

Ecological restoration is the process of assisting the recovery of ecosystems that have been damaged, degraded, or destroyed. One critical component of a successful restoration is the selection of an appropriate source population (*Kettenring et al., 2014*; *van Katwijk et al., 2016*), which can ensure population establishment and long-term success. Source selection may be determined by practical considerations such as cost and/or the availability of plants, but ideally it will be guided by ecological theory (*Montalvo et al., 1997*). *Kettenring et al. (2014)* recently highlighted three approaches for source selection: (1) planting from populations with preferred traits (the cultivar approach); (2) planting from local populations thought to be adapted to environmental conditions (the local adaptation approach); (3) planting genetically diverse mixtures (the genetic diversity approach). Although there is evidence in support of each of these approaches, few restoration efforts have explicitly tested their relative importance in contributing to successful initial establishment (*Kettenring et al., 2014*).

Seagrasses are an important foundation species that provide key ecosystem functions in coastal and estuarine systems. However, declines in the abundance and distribution of seagrasses along the world's developed coastlines threaten the health and sustainability of the systems that rely on these species (*Orth et al., 2006*). *Waycott et al. (2009)* estimated that a minimum of 29% of the known global extent in the distribution of seagrasses has been lost since 1879. Moreover, the International Union for Conservation of Nature (IUCN) Red List of Threatened Species recently classified 10 species of seagrass at elevated risk for extinction and three species as Endangered (*Short et al., 2011*). To mitigate for seagrass losses, considerable time and effort has been spent developing methodologies to improve restoration success rates, including models to facilitate the selection of transplant sites (e.g., *Short et al., 2002*; *Biber, Gallegos & Kenworthy, 2008*) and new planting techniques (e.g., *Calumpong & Fonseca, 2001*; *Lee & Park, 2008*; *Zhou et al., 2014*). Despite advances in these areas, restoration success rates are still low and variable (*Fonseca, Kenworthy & Thayer, 1998*; *Cunha et al., 2012*; *Bayraktarov et al., 2016*).

The majority of seagrass restorations to date have focused on eelgrass (*Zostera marina*), a species found in temperate waters along both coasts of the United States as well as throughout Europe and eastern Asia (*Green & Short, 2003*; *van Katwijk et al., 2016*). Eelgrass shows considerable physiological and morphological variation across populations and individual genotypes (*Hughes, Stachowicz & Williams, 2009*; *Tomas et al., 2011*; *Salo, Pedersen & Boström, 2014*; *Salo, Reusch & Boström, 2015*); thus, some populations may possess traits that make them more resilient to the transplanting process than others. There is some evidence that population-level plant traits serve a role in transplant success: *van Katwijk et al. (1998)* found that two of five *Zostera marina* populations failed to establish in a mesocosm transplant experiment and that morphology (shoot size) was one factor related to transplant success; likewise, *Lewis & Boyer (2014)* found differences in growth characteristics of two sources transplanted into a bare restoration site.
Current seagrass restoration guidelines suggest selecting source populations that are nearby to the transplant site and grow in comparable environments (i.e., are locally adapted to environmental conditions at the transplant site; *Addy, 1947*; *Calumpong & Fonseca, 2001*; *van Katwijk et al., 1998*; *van Katwijk et al., 2009*). Although there is evidence for local adaptation in seagrasses, few seagrass trials have explicitly tested the strength and consistency of this phenomenon as an approach for source selection. For example, *Hämmerli & Reusch (2003)* showed that eelgrass shoots produced significantly more biomass when transplanted back into their native site than when transplanted to a different location (i.e., home site advantage). In addition, *van Katwijk et al. (2016)* showed the proximity of source populations to restoration sites, often used as a proxy for similar environmental conditions, was positively correlated with the performance of seagrass restoration trials.

Given existing variation in functional traits in eelgrass (*Hughes, Stachowicz & Williams, 2009*; *Tomas et al., 2011*; *Salo, Reusch & Boström, 2015*), planting shoots from multiple source populations would seem like a reasonable approach to increasing transplant success (analogous to positive effects of genetic and species diversity (*Hughes et al., 2008*; *Cardinale et al., 2012*; *Kettenring et al., 2014*)). In fact, several studies with eelgrass have shown positive effects of within species genetic diversity on transplant success and productivity (*Williams, 2001*; *Hughes & Stachowicz, 2004*, *2011*; *Reusch et al., 2005*; *Reynolds et al., 2012*). Most of these experiments used individual plants of known genetic identity; because of the expense and time involved, this level of detail is generally not feasible for restoration efforts. However, the number of source populations may serve as a proxy for genetic diversity, facilitating transplant success. For example, *Reynolds et al. (2012)* conducted a seed-based restoration experiment testing the performance of plots created with seeds from single vs. multiple source sites, demonstrating a positive relationship between resulting plot-level genetic diversity (measured as allelic richness) and plant survival, production, and habitat provision. Whether source diversity of adult transplants has similar positive effects on transplant success has not yet been tested.

In this paper, we examined the effects of source identity and diversity (one vs. three source populations in a plot) on eelgrass transplant success over the course of one year. Our experiment was replicated at two sites that also served as source sites to test for local adaptation (home site advantage) and the generality of diversity effects. Morphological and genetic data were also collected to characterize variation within and among source populations and relate it to transplant performance. We hypothesized that both home site advantage and source diversity would have a positive effect on eelgrass shoot density.

## METHODS

### Source populations

*Zostera marina* shoots were collected from four source populations in Massachusetts with a range of environmental conditions (Table 1; Fig. 1). Approximately 500 vegetative shoots were haphazardly collected at a depth of 1.5 m mean low water along a 50 m transect oriented parallel to shore from each of the four locations (source sites) in November 2013. Shoots were removed by uprooting 3–5 cm of the rhizome and snapping

**Table 1 Environmental conditions at source population sites for 2013–2014.**

| Site | Tidal range (m) | Wave exposure | Temperature (°C) | Salinity (psu) | Substrate | Total nitrogen (μm) | Comments |
|------|-----------------|---------------|------------------|----------------|-----------|---------------------|----------|
| Dorothy Cove, Nahant | 3–4 | Moderate | −1–21 | 30–32 | Sand < 2% OM | 25–34 | Storm Water pipe at site |
| East Harbor, North Truro | 0.3 | Low | 0–29 | 20–30 | Sand < 2% OM | 20–47 | Tidally restricted; shallow embayment; 1 (Ha) eelgrass bed established 2008 |
| Pleasant Bay, Orleans | 2 | Low | 0–30 | 28–32 | Sand < 2% OM | 28–34 | Low flushing; shallow embayment; thriving 40 (Ha) bed |
| Sea Gull Beach, West Yarmouth | 3 | High | −2–24 | 30–32 | Sand < 2% OM | 6–18 | Shifting substrate |

**Note:**
Temperature and salinity for Dorothy Cove is from the Boston Harbor NDBC NOAA buoy; temperature and salinity data for East Harbor and Pleasant Bay is courtesy of Cape Cod National Seashore-National Park Service (CCNS-NPS); temperature and salinity for Sea Gull Beach, West Yarmouth was assumed to be similar as the Menauhant NERRS buoy; nutrient information for Dorothy Cove is courtesy of Center for Coastal Studies; nutrient information for East Harbor, Pleasant Bay, and Sea Gull Beach courtesy of CCNS-NPS, Pleasant Bay Alliance, and the Massachusetts Estuaries' Project (Dates 2012–2014).

the rhizome at the base of plants. Harvested eelgrass was cleaned of epiphytes and immediately stored in aerated or flowing seawater for less than 72 h before being transplanted at sites (*Davis & Short, 1997*).

## Morphological variation

Morphological variation was characterized within and across our source populations. The following traits were measured on 25 representative shoots collected from each source site that were not used in our test-transplanting experiment: the number of leaves per shoot, the above ground and below ground wet weight of each shoot, the distance between nodes on the rhizome (internode length), and the length, width, sheath length, and weight of the youngest fully mature leaf. We also measured leaf length on shoots in each plot during the experiment at six months post-transplantation.

## Genetic variation

Genetic variation within and among our source populations was characterized by analyzing DNA microsatellite variation for a subset of 25 shoots collected from each site. We extracted genomic DNA from ~200 mg of silica-dried leaf tissue using E.N.Z.A.® SP Plant DNA kit, following manufacturer's protocol (Omega Bio-Tek, Inc., Norcross, GA, USA). Nine microsatellite loci were amplified via three multiplex PCRs for each sample (see Table S1). All PCRs were run with the following conditions: 5 min denaturation at 95 °C, followed by 28 cycles of 90 s annealing at 60 °C, 30 s extension at 72 °C, and 30 s denaturation 95 °C, followed by a terminal extension of 30 min. Products were checked on 1% agarose gels before sending for fragment analysis via capillary electrophoresis at the DNA Analysis Facility at Yale University.

## Test-transplanting field experiment

Harvested eelgrass shoots were used in a field experiment and monitored for 11 months to determine the effects of source population identity (Nahant, East Harbor (EH), Pleasant Bay (PB), West Yarmouth; Fig. 1) and diversity (one vs. three source populations

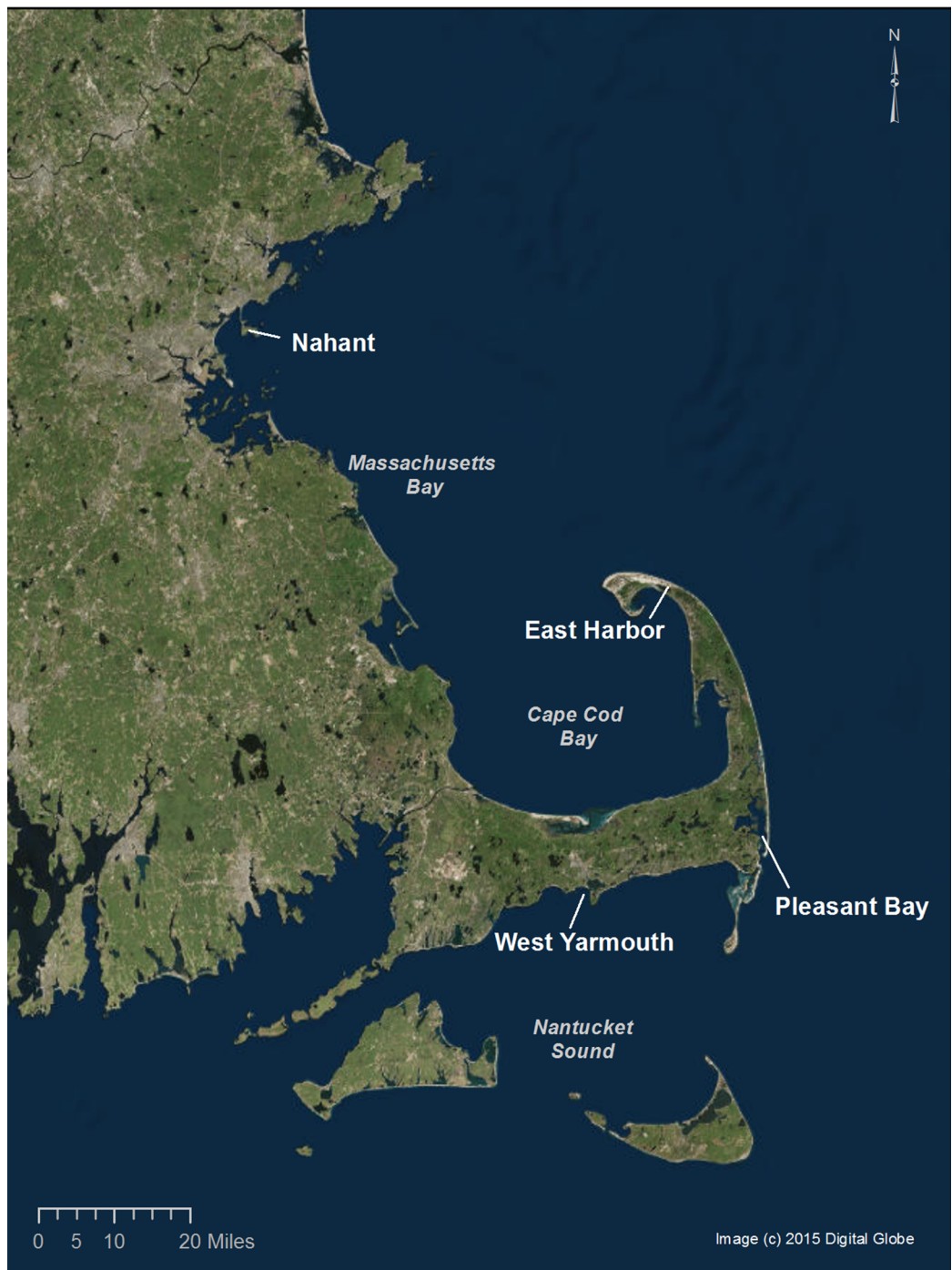

**Figure 1 Location of the source and transplant sites.** Note: East Harbor and Pleasant Bay served as source and transplant sites. Distances between sites are denoted in Table S2. Source: 2015 DigitalGlobe.

in a plot) on eelgrass performance (measured as shoot density). The experiment was replicated at two sites (EH and PB; Table 1; Fig. 1 and Fig. S1) to test for evidence of local adaptation and the generality of diversity effects. At EH, our experiment was established approximately ~200 m from the closest natural seagrass bed, whereas at PB we used large bare patches within the natural bed. We used a modified version of the TERF restoration

technique (*Fonseca, Kenworthy & Thayer, 1998*; *Short, Short & Burdick-Whitney, 2002*; *Leschen, Kessler & Estrella, 2009*) to maximize the relevance of our results for future efforts. Shoots were attached with jute string to the cross-hairs of gridded frames (60 cm × 60 cm; two shoots per cross-hair; 30 shoots per frame) made of PVC and jute string. Single source treatments contained 30 shoots from one of the four source populations, whereas multiple source treatments contained 10 shoots from three randomly selected source populations out of the four source populations. Treatments were assigned to plots in a complete randomized block design, with each block containing the four unique single source treatments and the four unique multiple source treatments. There were four blocks at each site, resulting in 32 plots per site (Fig. S1). Field experiments were approved by the National Park Service (Permit Number CACO-2013-SCI_0026).

All shoots in each plot were counted at each transplant site at four months (March) and then monthly from months six through 11 (May–November). Shoots did not expand outside of each plot and we were able to reliably assign shoots to cross-hairs until June (seven months post-transplantation) for within-plot diversity analyses (see below).

## Data analyses

Morphological variation across source populations prior to the initiation of our experiment was analyzed using a multivariate analysis of variance (MANOVA). The MANOVA indicated significant differences among source populations as well as multicollinearity. One dependent variable (leaf weight) had a partial correlation above 0.7 with leaf width and length, and was removed before conducting individual analyses of variance (ANOVA) on the remaining dependent variables. Tukey's HSD post hoc multiple comparisons tests were also performed to identify differences among source populations. The morphology dataset met the assumptions of the Kolmogorov–Smirnov test of normality and the Brown–Forsythe test of equal variance.

Morphological variation in leaf length was assessed at six months post-transplantation. To remove covariance due to non-independence within plots, we calculated the mean length of shoots for each source population at each diversity level. We analyzed both transplant sites together using a fully crossed three-factor ANOVA to test whether mean length of shoots varied with transplant site, source population, source diversity, or their interactions.

Microsatellite alleles were scored by eye using GENEMARKER V2.6.3 (SoftGenetics, Inc.), and only ramets for which all nine markers were successfully amplified and scored were used in further analysis ($n = 22$–23 per site). To identify genetically unique individuals and assign ramets to genets, we used GENECLONE 2.0 (*Arnaud-Haond & Belkhir, 2007*). We measured genotypic or clonal richness ($R$) as $R = (G–1)/(N–1)$, where $G$ is the number of unique genotypes and $N$ is the total number of shoots analyzed. We also used GENECLONE 2.0 to calculate two other measures of genotypic diversity, the Shannon index (H′) and clonal evenness (ED*). We then removed all repeatedly sampled genets, restricting the data set to unique multilocus genotypes, and used GENODIVE (*Meirmans & Van Tienderen, 2004*) to estimate allelic richness, effective number of alleles, expected and observed heterozygosity, and to calculate $F$-statistics.

The influence of source identity and diversity (single source vs. multiple source) on shoot density of the transplanted eelgrass was assessed by analyzing each transplant site separately, using two separate repeated measures ANOVAs (univariate split-plot approach) on the total number of shoots per plot, with block as a random effect and including all sampling dates (March, May–October). When the interaction between time and treatment (either source identity or source diversity) was significant, we conducted multiple one-way ANOVAs for each sampling date at each site, with treatment as a fixed effect and block as a random factor for months with equal sample sizes. Tukey's HSD post hoc multiple comparisons were performed to identify differences among source treatments. Local adaptation would be indicated if source sites perform better at 'home' than 'away' (e.g., if PB has higher density at PB than at EH), or if 'local' sources do better than 'foreign' sources (e.g., if PB has higher density at PB than other source sites).

When the ANOVA indicated a significant effect of source diversity, we calculated the net diversity effect by subtracting the expected multiple source density (calculated as the sum of each component single source divided by 3) from the observed multiple source density for each block (*Loreau & Hector, 2001*). We considered the net diversity effect significant if the 95% confidence intervals did not contain 0. We also conducted a two-factor ANOVA using the lme4 package in R (Version 3.2.3, R Foundation for Statistical Computing) to test whether the net diversity effect varied by source identity, by experimental site, or by their interaction.

The number of shoots per cross-hair collected the first three census periods (four months, six months, and seven months post-transplantation) was analyzed to assess whether shoots from the four source populations differed in their response to within-plot diversity. Covariance due to non-independence of cross-hairs within plots was removed by calculating the mean number of shoots per cross-hair for each source population within each plot; then we calculated the mean for each source population at each diversity level within a given spatial block, thus yielding a single value per spatial block for each source and diversity combination. We analyzed both transplant sites together, using a fully crossed three-factor ANOVA for each of the three census dates to test whether individual shoot survival varied with transplant site, source population, source diversity, or any of their interactions.

Statistical analyses were performed using JMP (Version 10.0.2, SAS Institute Inc.) unless otherwise noted.

## RESULTS

### Morphological variation

MANOVA tests of the variation in eelgrass shoot morphology among source populations at the time of transplantation using the Wilks' Lambda criteria were statistically significant ($F_{[18, 257.87]} = 20.60$, $p < 0.0001$). Subsequent univariate comparisons showed significant differences among source populations for all traits (Fig. 2; one-factor ANOVA: leaf width, $F_{[3,96]} = 12.85$, $p < 0.0001$; leaf length, $F_{[3,96]} = 18.50$, $p < 0.0001$; sheath length, $F_{[3,96]} = 49.40$, $p < 0.0001$; internode length, $F_{[3,96]} = 14.01$, $p < 0.0001$; leaves/shoot, $F_{[3,96]} = 22.79$, $p < 0.0001$; above/below ground weight, $F_{[3,96]} = 12.84$, $p < 0.0001$).

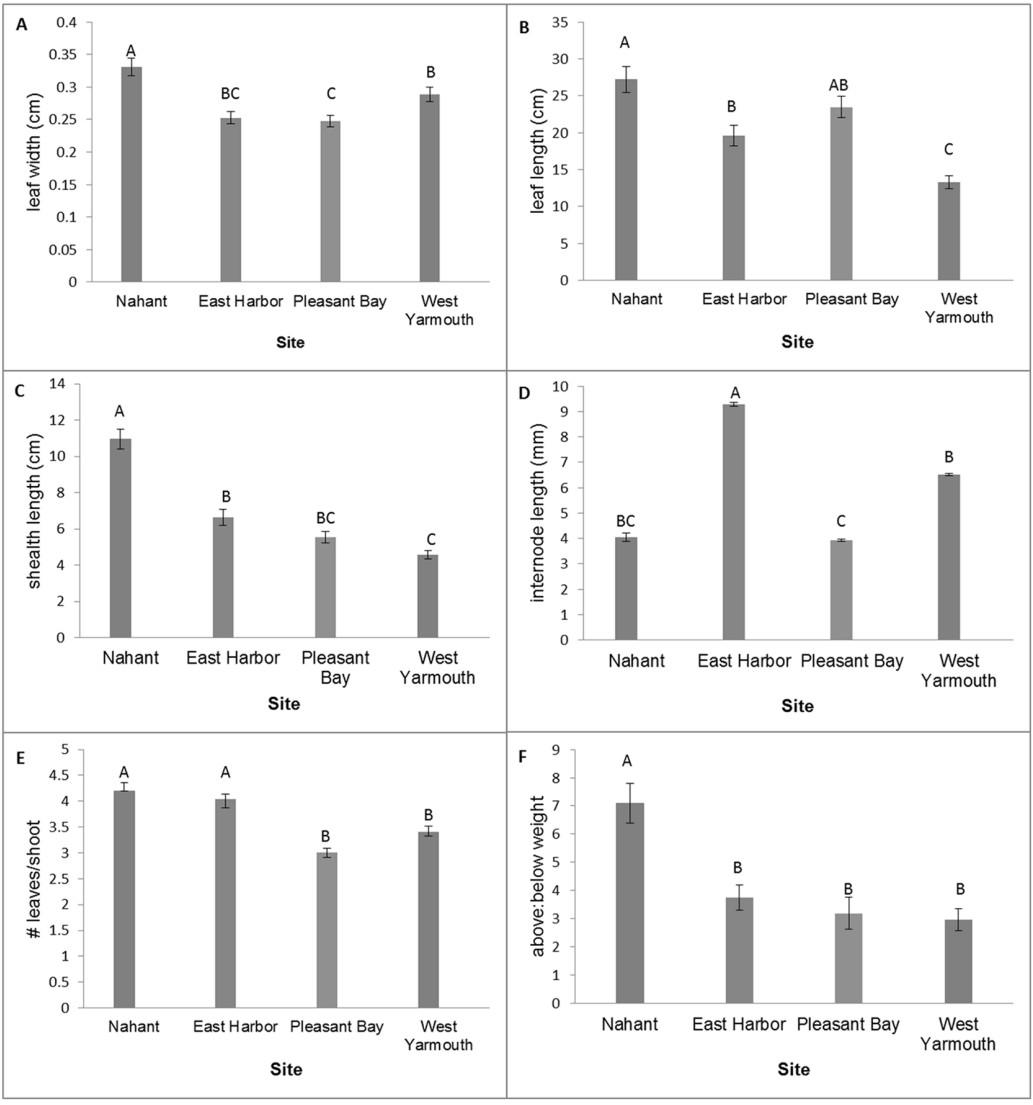

**Figure 2 Differences in shoot morphology among the four source populations at start of field experiment.** Significant differences were observed among source populations at $p < 0.05$ (mean ± SE); $n = 25$; Tukey's results denoted by *letters (A, B, C)*.

Nahant shoots had wider leaves, longer leaves (equivalent to PB shoots) and sheaths, and a greater ratio of above ground weight to below ground weight compared to shoots from other sites. West Yarmouth shoots had the shortest leaves and sheaths (equivalent to PB). EH shoots had the longest internode lengths while PB shoots had the shortest. Both Nahant and EH shoots had more leaves per shoot than shoots collected from West Yarmouth or PB (Fig. 2).

Leaf length was greater for shoots from Nahant (equivalent to EH) than shoots from PB or West Yarmouth at 6 months post-transplantation. This pattern was consistent across both transplant sites and diversity plots and there were no interactive effects observed between source, diversity, and/or site (three factor ANOVA: source, $F_{[3,419.80]} = 4.49$, $p = 0.0077$; site, $F_{[1,0.20]} = 0.01$, $p = 0.9356$; diversity, $F_{[1,43.58]} = 1.40$, $p = 0.2430$;
**Table 2 Indices of genetic diversity by source population.**

| Source population | N | R | H′ | ED* | $N_a$ | $N_e$ | $H_o$ | $H_e$ | $F_{is}$ |
|---|---|---|---|---|---|---|---|---|---|
| Nahant | 22 | 0.48 | 2.02 | 0.56 | 4.2 | 2.98 | 0.57 | 0.55 | −0.023 |
| East Harbor | 23 | 0.45 | 1.87 | 0.32 | 4.1 | 2.55 | 0.61 | 0.52 | −0.174 |
| Pleasant Bay | 23 | 0.82 | 2.87 | 0.66 | 6.1 | 3.72 | 0.71 | 0.65 | −0.090 |
| West Yarmouth | 23 | 0.82 | 2.89 | 0.79 | 5.1 | 2.54 | 0.58 | 0.53 | −0.096 |

**Note:**
N, number of genotyped shoots; R, clonal richness, calculated as (G−1)/(N−1), where G is the number of unique MLGs sampled; H′, Shannon index of clonal diversity; ED*, Simpson's evenness index (clonal evenness); $N_a$, mean number of alleles per loci; $N_e$, effective number of alleles per loci; $H_o$, observed heterozygosity; $H_e$, expected heterozygosity; $F_{is}$, inbreeding coefficient.

site × diversity = $F_{[1,3.14]} = 0.10$, $p = 0.7521$; site × source, $F_{[3,18.99]} = 0.20$, $p = 0.8936$; source × diversity, $F_{[3,75.62]} = 0.80$, $p = 0.4953$; source × diversity × site, $F_{[3,48.98]} = 0.52$, $p = 0.6678$.

## Genetic variation

Complete genotypes were obtained for 91 of the 100 shoots analyzed. All nine microsatellite loci used in this study were polymorphic, ranging from two alleles for CT-12 to 11 alleles for ZMC-12075. There were 60 unique genotypes found; none were shared across populations. Both clonal and allelic richness were higher in samples collected from West Yarmouth and PB than the other two source populations (Table 2). However, West Yarmouth and PB samples differed in the effective number of alleles per locus ($N_e$), which is the number of equally frequent alleles it would take to achieve a given level of gene diversity. PB samples possessed the highest $N_e$ of the four source populations tested here, and West Yarmouth the lowest, indicating slightly lower heterozygosity and less even distribution of alleles (Table 2). Pairwise comparisons of population differentiation detected significant structure between all pairs of source populations ($F_{st}$, $p < 0.001$), with the greatest differentiation found between West Yarmouth and EH (Table S2).

## Shoot density

Transplants grew and expanded until six months post-transplantation, however shoot density steadily declined at both sites from seven months post-transplantation to the end of monitoring (11 months; Fig. 3). When we assessed performance of source populations in single source plots we found an interaction between source population and time on shoot density in EH (repeated measures ANOVA: source, $F_{3,12.6} = 9.26$, $p = 0.0017$; time, $F_{6,64.44} = 99.39$, $p < 0.0001$; time × source, $F_{18,64.3} = 3.81$, $p = 0.0001$): shoot density was similar across sources sites for all months except March and May, when source plots containing shoots from Nahant or EH and Nahant or PB, respectively, exhibited the greatest shoot densities and West Yarmouth the lowest (Table S3; Fig. 3). There was no evidence for home site advantage, as EH shoots did not outperform other source sites in EH. At PB, all single source plots performed similarly, again demonstrating no home site advantage. We observed a significant effect of time, with shoot density gradually declining over the course of the experiment (repeated

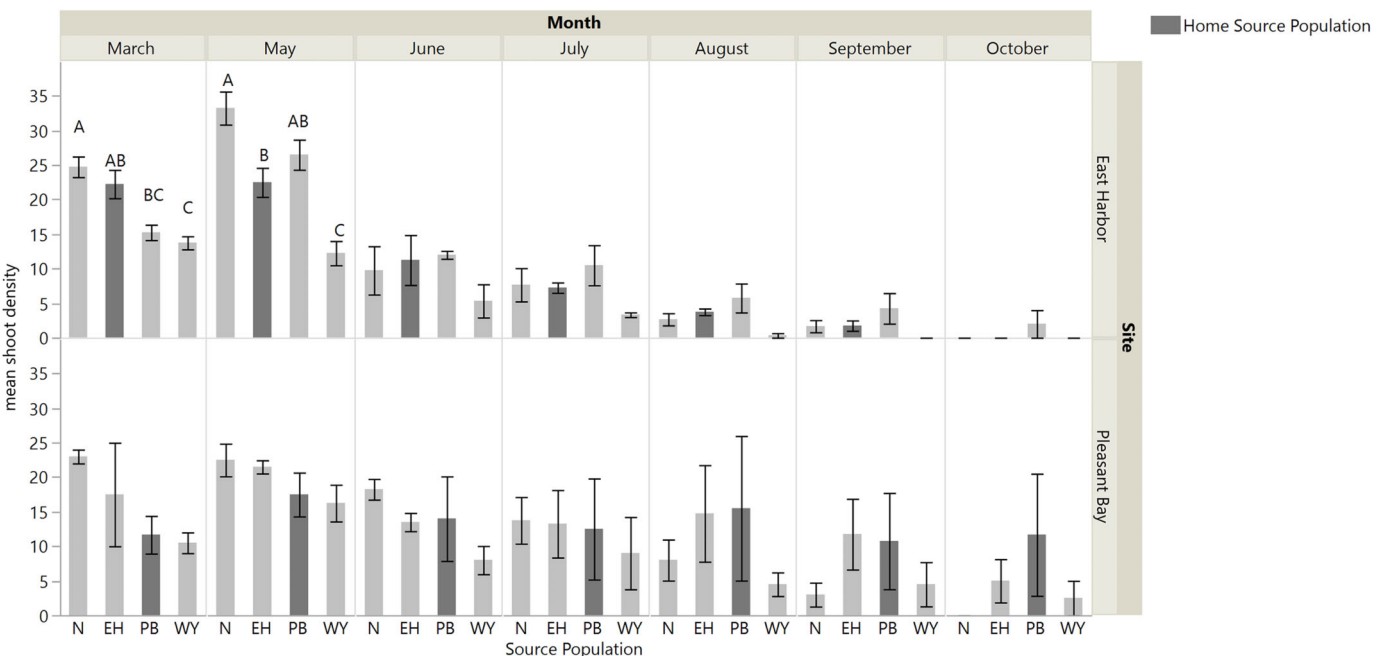

**Figure 3 Shoot density for each single source plot at each site (Nahant (N); East Harbor (EH); Pleasant Bay (PB); West Yarmouth (WY)).** Significant differences were observed among treatments in March and May at East Harbor at $p < 0.05$ (mean ± SE); $n = 3$–4; Tukey's results denoted by *letters (A, B, C)*. Home source populations shaded dark gray for each site.

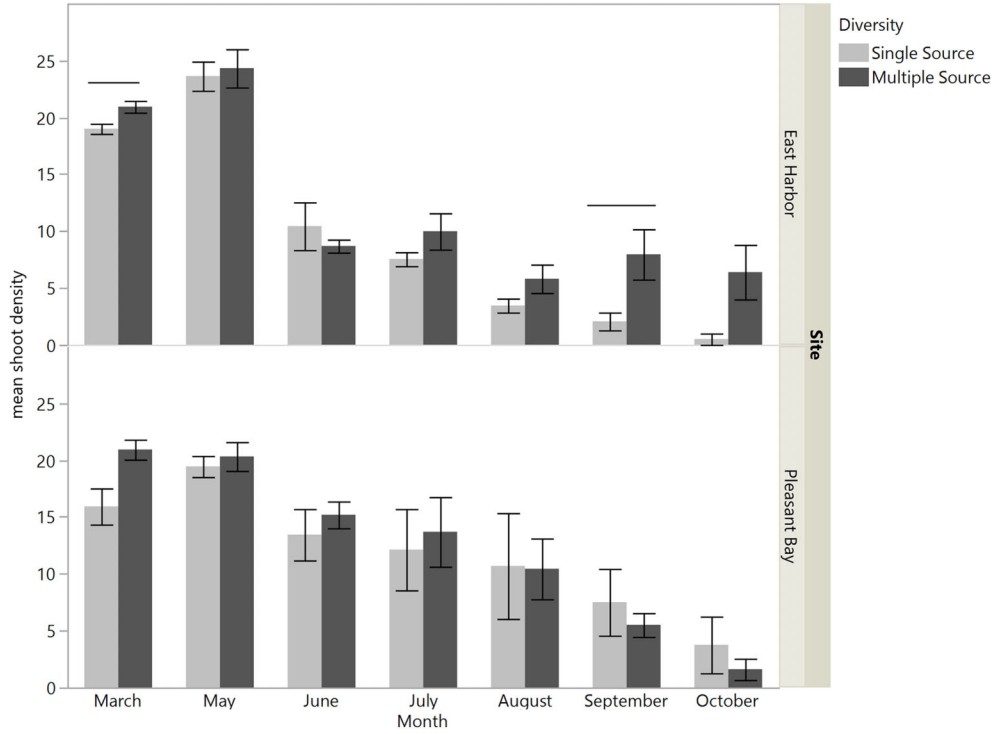

**Figure 4 Mean shoot density values per plot at each site for single source and multiple source plots for the months March and May through October.** Significant differences were observed between treatments only at East Harbor (denoted by solid bar) at $p < 0.05$ (mean ± SE); $n = 4$ for all months at each site except in March at Pleasant Bay (where $n = 3$).

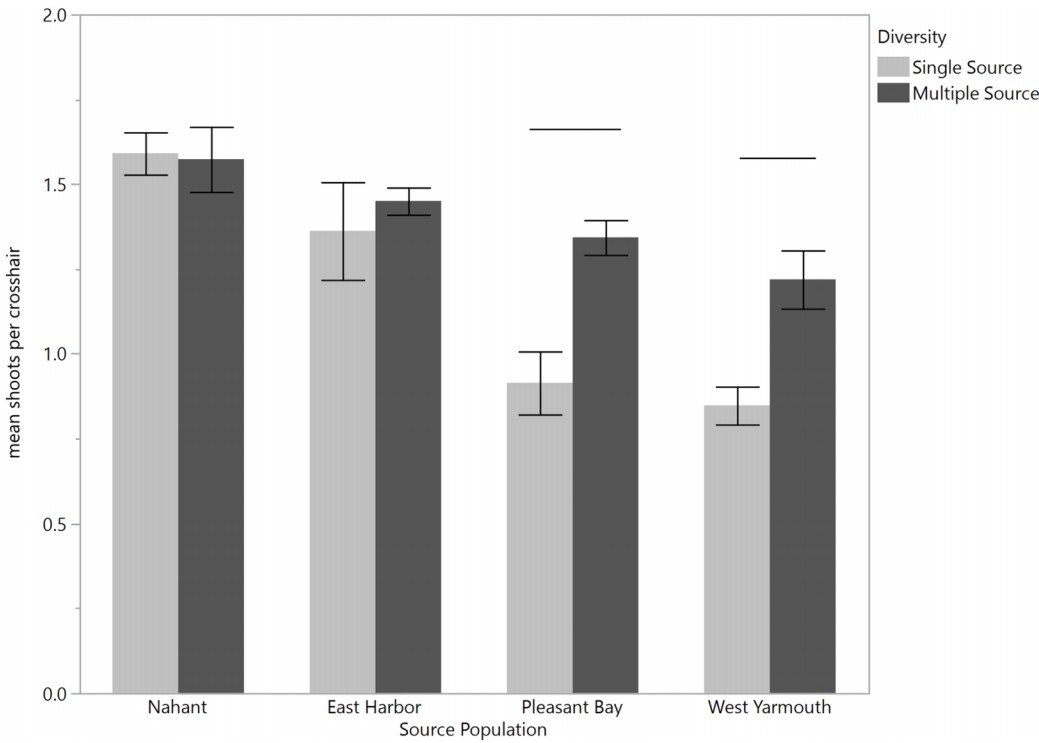

**Figure 5 Effect of source population and source diversity on mean number of shoots per cross-hair 4 months after transplantation (March).** Both transplant sites are pooled. Solid bars indicate significant differences in shoot performance between single source treatments and multiple source treatments (a priori contrasts, $p < 0.05$; mean $\pm$ SE).

measures ANOVA: source, $F_{3,12.08} = 0.62$, $p < 0.6155$; time, $F_{6,61.2} = 10.25$, $p < 0.0001$; time $\times$ source, $F_{18,61.2} = 1.20$, $p = 0.3022$).

There was an interaction between time and source diversity at EH when we compared the performance of single source and multiple source plots (Fig. 4; repeated measures ANOVA: diversity, $F_{1,6} = 5.87$, $p = 0.0516$; time, $F_{6,36} = 83.53$, $p < 0.0001$; time $\times$ diversity, $F_{6,36} = 2.40$, $p = 0.0468$), but only an effect of time at PB (Fig. 4; repeated measures ANOVA: diversity, $F_{1,5.89} = 0.014$, $p = 0.9094$; time, $F_{6,32.98} = 25.94$, $p < 0.0001$; time $\times$ diversity, $F_{6,32.98} = 0.56$, $p = 0.7519$). At EH, shoot density was higher for multiple source plots compared to single source plots in March (four months post-transplantation) and September (ten month post-transplantation; Table S3; Fig. 4). The net diversity effect analysis corroborated these results: there was a positive overall effect of diversity in March (mean $\pm$ 95% CI = 3.25 + 1.14), but this positive effect was only significant at EH (site $F_{1,20} = 8.71$, $p = 0.008$). In September, the overall diversity effect was not significant because a positive diversity effect in EH was counter-acted by a non-significant diversity effect in PB (site $F_{1,24} = 7.32$, $p = 0.01$).

When we assessed eelgrass performance at the scale of shoots per cross-hair (i.e., within grid), we found that in March the effect of source diversity depended on which source the eelgrass shoots came from (source $\times$ source diversity $F_{3,40} = 4.07$, $p = 0.01$; Fig. 5) in a pattern that was consistent across both transplant sites (source $\times$ source diversity $\times$ site $F_{3,40} = 1.66$, $p = 0.19$; Table 3A). Shoots from PB and West Yarmouth performed

**Table 3** Analysis of Variance on the mean number of shoots remaining per cross-hair at both sites in (A) March (four months), (B) May (six months), and (C) June (seven months post-transplantation).

| Source | df | S-of-S | F-ratio | p |
|---|---|---|---|---|
| **A.** | | | | |
| Site | 1 | 0.136 | 3.098 | 0.086 |
| Source population | 3 | 2.481 | 18.877 | **<0.0001** |
| Source diversity | 1 | 0.743 | 16.968 | **0.0002** |
| Site × source population | 3 | 0.140 | 1.062 | 0.376 |
| Site × source diversity | 1 | 0.154 | 3.522 | 0.068 |
| Source population × source diversity | 3 | 0.534 | 4.066 | **0.013** |
| Site × source pop'n × diversity | 3 | 0.218 | 1.661 | 0.191 |
| **B.** | | | | |
| Site | 1 | 0.828 | 8.983 | **0.005** |
| Source population | 3 | 5.601 | 20.267 | **<0.0001** |
| Source diversity | 1 | 0.0001 | 0.002 | 0.965 |
| Site × source population | 3 | 2.092 | 7.567 | **0.0003** |
| Site × source diversity | 1 | 0.044 | 0.482 | 0.491 |
| Source population × source diversity | 3 | 0.065 | 0.236 | 0.870 |
| Site × source pop'n × diversity | 3 | 0.557 | 2.013 | 0.126 |
| **C.** | | | | |
| Site | 1 | 1.942 | 24.717 | **<0.0001** |
| Source population | 3 | 0.783 | 3.321 | **0.027** |
| Source diversity | 1 | 0.062 | 0.791 | 0.378 |
| Site × source population | 3 | 0.989 | 4.195 | **0.010** |
| Site × source diversity | 1 | 0.296 | 3.762 | 0.058 |
| Source population × source diversity | 3 | 0.209 | 0.885 | 0.455 |
| Site × source pop'n × diversity | 3 | 0.186 | 0.788 | 0.506 |

significantly better in multiple source than single source plots (a priori contrasts, $p = 0.001$ and $p = 0.0003$), whereas shoots from Nahant and EH performed equivalently regardless of the diversity of neighboring shoots ($p > 0.1$). As with the grid-level analysis above, this pattern changed over time. By May and June, shoot performance depended largely on the interactive effects of source identity and site (source × site; May: $F_{3,44} = 7.57$, $p = 0.0003$; June: $F_{3,48} = 4.19$, $p = 0.01$), with no support for a home site advantage for transplants at either location or date (Fig. S3). No significant effect of source diversity was detected at the shoot level after the first census (Tables 3B and 3C), although in June there was a marginal effect that differed between the sites (diversity × site $F_{1,48} = 3.76$, $p = 0.06$), such that source diversity tended to have a positive effect on shoot density at PB but no effect at EH (Fig. S4).

## DISCUSSION

Our study examined the effects of seagrass source identity and diversity on *Zostera marina* transplant success in a field experiment that was replicated at two locations that also served as source sites to test for local adaptation and the generality of source diversity

effects on shoot density. We showed that transplants grew and expanded until six months post-transplantation, however, after that period shoot density steadily declined at both sites until the end of our experiment. Prior to declines, we found that source identity was important for early transplant performance at our EH site. Single source plots containing eelgrass from West Yarmouth had the lowest shoot densities, whereas plots containing eelgrass from Nahant and PB exhibited the greatest shoot densities at six months post-transplantation. In contrast, source identity did not significantly influence performance at the PB site. Thus, our results support the hypothesis that some seagrass populations may be more suitable sources for transplanting than others (*Meinesz et al., 1993*; *van Katwijk et al., 1998*), yet suitability will likely vary across sites.

There are several characteristics of source populations that could have contributed to the variation in performance we observed before our plots declined. Proximity of source populations to a restoration site has been linked to increased seagrass transplant success (*van Katwijk et al., 2016*), potentially due to similarity in environmental conditions among nearby sites and local adaptation of plants to these conditions. By transplanting two source populations within/adjacent to the source site, our experimental design allowed us to test for local adaptation of two source populations. We showed that shoots transplanted back to their home site did not consistently outperform shoots from other source sites (i.e., no local vs. foreign advantage), nor did shoots transplanted back to their home site outperform shoots from that same population planted in a different site (i.e., no home vs. away advantage). In addition, we found no clear benefits of shoots collected from nearby sites (e.g., EH shoots at the PB site) and/or a clear relationship between shoot performance of transplants and environmental conditions at source sites. Field experiments by *Piazzi et al. (1998)* also found no evidence for local adaptation in transplants and instead showed that *Posidonia oceanica* shoots from distant beds had higher rhizome growth and ramification than shoots from nearby source sites. Based on the results of our experiment and others, we suggest that source characteristics other than home site advantage and/or geographic proximity to the transplant site may have greater influence on initial transplant success. However, since we did not transplant Nahant or West Yarmouth populations back to their home sites we should not rule out the role of local adaptation in early transplant success. Moreover, potential mismatch between donor traits and the recipient site characteristics may yet become apparent over longer time scales than those of our study. For example, analysis of a *Zostera noltii* restoration experiment four years after transplantation revealed that the new seagrass patches that had appeared in the lagoon since the experiment were genetically distinct from the transplanted shoots, and likely recruited from remnant patches of the original population (*Jahnke et al., 2015*).

Morphological variation among seagrass populations can be due to environmental and/or genetic factors and has measureable effects on community composition, structure, and function (*Kuo & den Hartog, 2006*; *Hughes, Stachowicz & Williams, 2009*). In our study, we documented a diversity of morphologies among source populations, characterized by significant variation in internode length, leaf length, sheath length, and leaf width. Of our source sites, Nahant had on average the longest leaves both initially

as well as up to six months post-transplantation, and plots from Nahant had among the highest densities during this same time period. Alternatively, single source plots from West Yarmouth, the source population with the shortest leaves initially, performed the worst during this same time period. *van Katwijk et al. (1998)* also observed a relationship between shoot performance and morphology, however, the authors found that smaller-sized shoots (shoots with shorter and narrower leaves) outperformed larger shoots when grown in a controlled outdoor mesocosms over a 2-year period. Based on our results, we suggest that the utility of morphological traits as an indicator of source population performance, at least for initial post-transplant survival, should be further assessed.

Another criterion suggested for source site selection is the genetic diversity of the donor population. In clonal plants such as *Zostera marina*, genetic variation can be measured in different ways, including the diversity of genotypes (clones), and the diversity of genes (alleles) found in individuals in a population or plot. While these measures are correlated at the lowest levels of genetic variation (see *Massa et al., 2013*), both types of diversity may contribute independently to the positive relationship between seagrass genetic diversity and ecosystem function increasingly documented in manipulative studies (e.g., *Hughes & Stachowicz, 2004*, *2011*; *Reusch et al., 2005*; *Reynolds et al., 2012*). Heritable genetic variation in functional traits largely determines a population's evolutionary potential; thus planting individuals from a more diverse source may promote population persistence over a longer time frame (see, e.g., *Reynolds et al., 2012*). A third measure of genetic diversity, heterozygosity, or the diversity of alleles contained within an individual, may also indicate a good source population; heterozygosity may not reflect total allelic diversity at the population scale, but greater heterozygosity can have immediate benefits for the fitness of individual eelgrass ramets and population growth (*Williams, 2001*; *Hämmerli & Reusch, 2003*). In this study, we found no correlation between genetic or genotypic richness in our four source populations and their subsequent growth and survival when transplanted into single-source plots. Notably, samples from two of our four sources, West Yarmouth and PB, exhibited similarly high genotypic diversity, genotypic richness, and allelic richness, but differed in transplant performance: West Yarmouth showed the lowest performance of the four sources overall (across both transplant sites and over time), while shoots from PB faired better, significantly so for some sampling periods at EH. On the other hand, mean heterozygosity and its correlate, effective allelic richness, were lowest in the (poor performing) samples from West Yarmouth. However, heterozygosity was not a consistent predictor of source performance, at least over the range of genetic diversity we were able to test here.

The number of source populations used in restoration may function as a proxy for genetic, morphological and functional diversity of transplants. Although genetic diversity *within* each source population did not clearly predict transplant performance, we did detect a positive effect of increasing source population number at some points during the experiment. While the strength of this effect varied across experimental sites and through time, multiple source plots consistently performed as well or better than single-source plots (i.e., there was never a negative effect of source number). Our per cross-hair

analysis suggests that the positive effects of diversity were not simply the result of the presence of high-performing individual sources (also known as a sampling effect; *Hector, 1998*). Rather, the performance of otherwise weak-performing source populations (e.g., West Yarmouth) improved in multiple source plots. The positive effects of plot-level source diversity in our study are consistent with findings of increased production with higher plot-level genotypic richness in *Zostera marina* (*Hughes & Stachowicz, 2004*, *2011*; *Reusch et al., 2005*). Taken together, they suggest that it is genetic diversity at the plot scale (i.e., the scale at which individual plants interact with one another), and not at the population level, that is most likely to confer the expected benefits of biodiversity. Furthermore, prior experiments with *Zostera marina* have found that effects of genetic diversity increased through time (*Hughes & Stachowicz, 2011*; *Reynolds et al., 2012*). Thus, we suggest that restoration efforts consider planting eelgrass shoots from multiple sources in combination when feasible, rather than planting shoots from a single source, to increase the chances of transplant success. *Ort et al. (2012)* has also suggested using multiple populations (in close proximity) based on the population structure of *Zostera marina* in estuaries. However, of key concern would be the distance over which source populations should be combined, as mixing historically isolated populations can lead to negative consequences (e.g., outbreeding depression) if populations are sufficiently genetically differentiated (see review by *Hufford & Mazer, 2003*). Thus sources should be selected from within a relatively nearby area or, ideally, a defined management unit (*Olsen, Coyer & Chesney, 2014*). Given the overall low survival of transplants and relatively short time frame of this study, our results capture only asexual expansion/contraction of genets, not sexual reproduction. More importantly, while the significant pairwise $F_{st}$ values indicate genetic subdivision among the four source populations, a recent molecular study of eelgrass populations in southern New England and New York indicates that the more substantial break in population structure occurs south of Cape Cod, and that the four sites used here are part of the same putative metapopulation (*Short, Burdick & Moore, 2012*).

Shoot density at both experimental sites gradually declined through time after six months, with very low abundance one year post-transplanting. Several factors may have contributed to this decline. Water quality measurements recorded near our field experiments by CCNS-NPS indicate that by the end of May (six months post-transplantation) water temperatures in PB and EH consistently exceeded thresholds for optimal photosynthesis and growth (20–23.3 °C; *Lee, Park & Kim, 2007*; *Staehr & Borum, 2011*), with temperatures as high as 27 °C during the summer months. Thus, the high water temperatures could have contributed to the high mortality of eelgrass transplants during this period. In addition to high water temperatures, EH experienced a macroalgal bloom (*Cladophora* spp.) that appeared in July and persisted through September. The bloom carpeted the transplants, and undoubtedly limited the light available, thus limiting photosynthesis, but had no visible impact on the natural beds in the area. The PB site did not experience any algal blooms, however, disturbance by horseshoe crabs was observed in the area. Furthermore, the frames that we used for planting facilitated plant establishment initially, but they appeared to inhibit rhizome

development at later stages of the experiment, uprooting the plants at some cross hairs and causing shoot mortality. While frames have been shown to have a benefit during the initial stages of transplanting, recent reviews of seagrass restoration efforts have noted a decline in the benefit of frames through time and suggest removing frames following initial plant establishment (*van Katwijk et al., 2016*). Lastly, the relatively low number of shoots (<1,000) used at each of our sites may have hindered long-term plant establishment, as the magnitude of planting effort is a primary predictor of seagrass transplant success (*van Katwijk et al., 2016*).

The goal of this study was to explore how the identity and diversity of source populations may influence eelgrass transplant success. Although survival of all transplants was ultimately very low at both sites, we did find that early transplant success varied among source populations, and that the pattern of performance was inconsistent with local adaptation. This finding may reflect the short-time scale of our study, as local adaptation is predicted to be more important in long-term plant establishment (*Kettenring et al., 2014*; *Jahnke et al., 2015*). Our results support the importance of morphological characteristics of the source population (i.e., shoot size) in early transplant success (*van Katwijk et al., 2009*) suggesting that planting sources with particular traits (similar to the cultivar approach) may benefit rapid plant establishment (*Kettenring et al., 2014*). Although we did not explicitly manipulate the genotypic richness of source plants, we found limited evidence that population diversity increased individual or plot performance; this pattern was significant at one of the transplant sites for two dates of the seven sampled. Others have suggested using a multi-source approach to further ensure that restored eelgrass beds do not have reduced genetic diversity compared to natural beds (cf. *Williams & Davis, 1996*). Based on our overall findings that transplant success varied among source populations, and that population diversity at the plot level had positive but limited effects on individual and plot performance, we support further tests of the multi-source approach for restorations.

## ACKNOWLEDGEMENTS

We thank K. Benes, F. Choi, T. Hanley, T. McCadden, G. Pacela, T. Rogers, F. Schenck, A. Stanclift, A. Costa, R. Novak, J. Burkhart, and A. Thime for help in the field. We also thank J. Bruno and three reviewers for their comments.

### Funding

A Keene State College Faculty Development Grant was obtained to conduct the molecular analyses. The funders had no role in study design, data collection and analysis, decision to publish, or preparation of the manuscript.

### Competing Interests

The authors declare they have no competing interests.

## Author Contributions

- Alyssa B. Novak conceived and designed the experiments, performed the experiments, analyzed the data, contributed reagents/materials/analysis tools, wrote the paper, prepared figures and/or tables, reviewed drafts of the paper.
- Holly K. Plaisted conceived and designed the experiments, performed the experiments, contributed reagents/materials/analysis tools, reviewed drafts of the paper.
- Cynthia G. Hays conceived and designed the experiments, performed the experiments, analyzed the data, contributed reagents/materials/analysis tools, wrote the paper, prepared figures and/or tables, reviewed drafts of the paper.
- A. Randall Hughes conceived and designed the experiments, performed the experiments, analyzed the data, contributed reagents/materials/analysis tools, wrote the paper, reviewed drafts of the paper.

## Field Study Permissions

The following information was supplied relating to field study approvals (i.e., approving body and any reference numbers):

Field experiments were approved by the National Park Service (Permit Number CACO-2013-SCI_0026).

## Data Deposition

The raw data in this article is included in the manuscript in the tables and figures and has also been supplied as Supplementary Dataset Files.

## Supplemental Information

Supplemental information for this article can be found online at http://dx.doi.org/10.7717/peerj.2972#supplemental-information.

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
