# Peer review of "Limited effects of source population identity and number on seagrass transplant performance"

_PeerJ, doi:10.7717/peerj.2972_

## Round 0.1 · original submission · Major Revisions

I am very sorry for the delay in making this decision. It took a while and lots of emails to get two good reviews.

·

Basic reporting

No comments.

Experimental design

The article seems to be missing some details on the sites - were there differences in the environmental conditions aside from proximity to seagrass beds? I'd like to see that level of detail included and the similarities/differences considered in the discussion.

Validity of the findings

No comments.

Reviewer 2 ·

Basic reporting

The authors undertook a transplant experiment to evaluate alternative theories about the role of source population selection in seagrass restoration success. Specifically, they evaluated two approaches to source selection: (1) the hypothesized benefit of planting genetically diverse mixtures and (2) the hypothesized local adaptation approach (home site advantage) to confer restoration success. These two hypotheses were evaluated using blocked transplant experiment at two restoration sites using sources from 4 extant seagrass populations.


This manuscript needs a more careful edit before the next submission to correct layout inconsistencies, spelling mistakes, etc.
- paragraphs are inconsistently indented (i.e. Line 121, 129....)
-Axis labels are inconsistently capitalized (Fig 2: "leaf width" vs "Above/below weight)
-spelling errors (Line 129 "varaiation")
-inconsistent abbreviations of units (Line 123 "m" and "meters" within the same sentence!; Line 142-3 includes "s" and "seconds"---are these denoting the same units?)
-terminology consistency: Line 134 introduced "leaf length" but the discussion refers to "shoot length"---are these the same? if so, keep the phrasing consistent)
-numeric references are inconsistent ("four" vs "4"; Line 160 and Line 161)
-comma needed after "plots" in Line 178
-spell out acronyms when first used (Line 202 "PB")

Experimental design

The experimental design is sound and appropriately presented. A few specific questions:

-shoot density is listed in the methods as the sole response variable for evaluating transplant performance, but leaf length is also measured and analyzed in the methods and results. Why is leaf length (Line 135) not listed as a response variable for performance (Line 150)?

-why was growth rate not considered as a indicator of transplant performance?

-How might transplant location (i.e. within large bare patches in the extant bed versus adjacent to extent bed) within the experimental sites impacted your results?

-Where is AE (Line 225) in Table 1. Is AE = Na?

-Why was the diversity effect analysis (Line 283-288) not conducted separately for each site (as the repeated measured ANOVA was?) Especially given the differences observed between sites (Line 286-288)?

-Consider moving Fig S2 and Table S5 into main text. Not clear why this figure is in the supplemental material as within plot diversity is a main component of the discussion. Would be easier to follow the analysis output in Table and Figure form, than text.

Validity of the findings

While the theoretical context was compellingly laid out in the introduction, the paper lacks clarity to readily allow the reader to evaluate the outcome of two hypotheses. The discussion convincingly concluded that geographic proximity of the source population (i.e. H2; local advantage and home advantage) was not found to be a strong predictor of restoration success. However, the reader is left to wonder whether the conclusions drawn from the discussion with regards to H1 (genetic diversity) are truly reflective of genetic differences (H1; Lines 380-392) or the morphological differences among source populations so clearly established early in the results (Fig 2; Lines 342-354). The positive effects of plot-level source diversity (Line 380-409) are currently ascribed to genetic diversity and incorporated into recommendations for restoration to favor genetic diversity (Lines 395-403). However, the authors fail to address that the morphological differences in internode length, shoot length, shoot width etc. are inextricably linked (or confounded) with the genetic differences detected among source populations. Hence, the argument weighing genetic diversity over morphological diversity (i.e. the cultivar approach) cannot be convincingly made ---and this limitation should be acknowledged in the discussion. More clearly stating that this experiment could not distinguish between these two theoretical approaches presented in the introduction (Lines 54-58), would clarify the findings of the study.

Overall recommendations therefore are to:
- revise the discussion in light of the limitations of the experimental results (confounded genetic and morphological differences).
- Table S1 could be moved to the main text to allow the reader to see more clearly the environmental variation between the source sites that may be influencing the morphological differences
-Likewise, moving Fig S2 and Table S5 into main text would improve the readers ability to assess your findings.

Reviewer 3 ·

Basic reporting

The ms is well-written and well-cited. Figures are clear and appropriate. I would say the ms meets the standards of the journal.

Experimental design

Experimental design is appropriate and the study was rigorous. Research question is relevant and meaningful.

Validity of the findings

My only issue is that the authors need to temper a few of their conclusions and to be clear the extent to which they can be drawn from the results. The authors should tone down their recommendation to plant from multiple sources to “increase the chances of transplant success” (line 397 in the discussion). This study only showed an advantage of polycultures on a couple of dates out of the seven measured and at only one of the transplant sites. This is at most suggestive of a limited advantage of polycultures and should be stated so explicitly. The abstract stating that polycultures performed similarly or better than monocultures should be more explicit that this was at one of two sites, while at the other site, there was no benefit and in fact there was a negative effect of polyculture over time. Line 443 should read something like, “we found limited evidence that population diversity at the plot-level increased both individual and plot performance, as this pattern was significant on only XX dates of the seven in the study and at only one of the transplant sites, while at the other….(see comment above on language for the abstract)”. Others have suggested using multiple sources at restoration sites to bet hedge; the data here only very weakly support this and further study is needed.

On the role of morphology in predicting transplant success, in the discussion there is the strong statement of the “importance of morphological characteristics of the source population (e.g., shoot size) in early transplant success (van Katwijk et al., 2009) suggesting that planting sources with particular traits (similar to the cultivar approach) can benefit rapid plant establishment (Kettenring et al., 2014)”. First of all, the fact that there were no differences by source by 7 months suggests that the starting morphological characteristics were decidedly not important in determining transplant success. Further, the authors should revisit the part of the intro (line 80) that mentions a previous study finding leaf width was a key predictor; how do this study’s results compare? The authors have focused on leaf length as the morphological characteristic that might be predictive, but they should point out the caveats of their findings in the abstract and discussion: Pleasant Bay should have been a good choice along with Nahant (and PB did look like a good choice at 6 months, but not at 4 and not after 6). Don’t overstate and this will be a great contribution!

---

## Round 0.2 · accepted · Accept

I appreciate the edits you've made in response to the reviewer suggestions. The only further revision I suggest is that the axes labels on most of your graphics are pretty small - I'd increase the font size so that when the figures are shrunken they are still readable.